# Exosomal Non Coding RNA in LIQUID Biopsies as a Promising Biomarker for Colorectal Cancer

**DOI:** 10.3390/ijms21041398

**Published:** 2020-02-19

**Authors:** Amro Baassiri, Farah Nassar, Deborah Mukherji, Ali Shamseddine, Rihab Nasr, Sally Temraz

**Affiliations:** 1Department of Anatomy, Cell Biology and Physiology, American University of Beirut Medical Center, Riad El Solh, Beirut 1107, Lebanon; asb11@mail.aub.edu; 2Department of Internal Medicine, Hematology/Oncology division, American University of Beirut Medical Center, Riad El Solh, Beirut 1107, Lebanon; fn17@aub.edu.lb (F.N.); dm25@aub.edu.lb (D.M.); as04@aub.edu.lb (A.S.)

**Keywords:** colorectal cancer, exosome, liquid biopsy, miRNA, lncRNA, circRNA

## Abstract

Colorectal cancer (CRC) is one of the most common cancers worldwide, with a high mortality rate, especially in those that are diagnosed in late stages of the disease. The current screening blood-based markers, such as carcinoembryonic antigen (CEA) and carbohydrate antigen 19-9 (CA19-9), have low sensitivity and specificity. Meanwhile, other modalities are either expensive or invasive. Therefore, recent research has shifted towards a minimally invasive test, namely, liquid biopsy. Exosomes are favorable molecules sought in blood samples, since they are abundant, stable in circulation, and harbor genetic information and other biomolecules that could serve as biomarkers or even therapeutic targets. Furthermore, exosomal noncoding RNAs, such as miRNAs, lncRNAs, and circRNAs, have demonstrated the diagnostic potential to detect CRC at an early stage with a higher sensitivity and specificity than CEA and CA19-9 alone. Moreover, they have prognostic potential that is TNM stage specific and could serve as predictive biomarkers for the most common chemotherapeutic drug and combination regimen in CRC, which are 5-FU and FOLFOX, respectively. Therefore, in this review, we focus on the role of these exosomal noncoding RNAs as diagnostic, prognostic, and predictive biomarkers. In addition, we discuss the advantages and challenges of exosomes as a liquid biopsy target.

## 1. Introduction

Recent research has shifted into developing minimally invasive biomarkers in the form of liquid biopsy, which is the sampling and analysis of various types of cells and molecules collected from biological fluids. The sampled fluid may be in the form of blood, plasma, cerebrospinal fluid, bronchoalveolar lavage fluid, pleural effusions, saliva, or urine. The diverse cells and molecules analyzed include circulating tumor cells (CTCs), circulating free DNA (cfDNA), circulating tumor DNA (ctDNA), circulating messenger RNAs (mRNAs), micro-RNAs (miRNAs), long non-coding RNAs (lncRNAs), circular RNAs (circRNAs), tumor-educated platelets, proteins, peptides, metabolites, and exosomes. The eclectic information that can be acquired provides liquid biopsy with vast clinical potential via the identification of diagnostic, prognostic, and predictive markers of different diseases, including cancer. Diagnostic biomarkers facilitate cancer screening and tumor heterogeneity detection. Prognostic biomarkers allow for risk estimation for progression versus relapse. The predictive biomarkers can be utilized to identify therapeutic targets, detect drug-resistance, and monitor response to treatment [1]. Across several cancers, evidence of the pivotal role that liquid biopsies could play in patients’ management has been growing, especially in colorectal cancer (CRC).

Several types of liquid biopsies have already demonstrated their potential role in CRC diagnosis, prognosis, and therapy prediction. For CRC diagnosis, a study has demonstrated that tumor heterogeneity could be detected in blood via ctDNA with 97% accuracy when correlated to corresponding tissue biopsy results. Moreover, in some cases, it might be better than tissue biopsy [2]. Furthermore, we can potentially diagnose patients with KRAS-mutant CRC via a liquid biopsy, since KRAS mutant fragments are detected in patients’ ctDNA [3]. Currently, there is an FDA approved screening tool that detects the SEPT9 promotor region methylation in plasma, which is considered to be a specific biomarker of early CRC stages. Across several studies, it has demonstrated good sensitivity and specificity as a diagnostic tool for CRC [4]. Some prognostic biomarkers include total cfDNA levels that correlate with DFS and OS, irrespective of tumor stage, use of adjuvant chemotherapy, tumor marker, and sample type [5,6,7,8,9,10,11,12,13]. Additionally, it has been demonstrated that ctDNA is a more accurate predictor of relapse than CEA [14]. Another prognostic biomarker is cell-free microRNA (cf-miR), such as cf-miR-21, 203, or 1290, whose high expression is associated with poor prognosis [15,16,17]. In addition, the high expression of cf-miR-200c in serum is associated with lymph node and distant metastasis [18]. Furthermore, CTC-positivity is associated with decreased disease-free survival (DFS) and poor overall survival (OS), irrespective of stage [19,20,21,22,23,24,25,26]. Liquid biopsies have also been able to identify mutations that cause resistance to EGFR inhibitors [27,28]. We are particularly interested in exosomes, in addition to all of these biomarkers.

CRC is the third most common cancer worldwide and it ranks second in terms of mortality [29]. Five-year survival rates for diagnosis at early stages is as high as 90% in comparison to the dismal rate of 13% for stage IV. Screening tests available today either have low sensitivity and specificity, high cost, or are invasive in nature, which affects patients’ compliance. As such, there is a need to develop robust, inexpensive, and minimally invasive screening biomarkers to detect CRC at an early stage [30]. Currently, the two blood-based biomarkers commonly utilized, Carcinoembryonic antigen (CEA) and carbohydrate antigen 19-9 (CA19-9), are not effective due to their insufficient sensitivities and specificities [31,32]. Hence, we ought to find more reliable molecules to study should liquid biopsies be sought. The plethora and stable nature of exosomes in circulation make them a potentially favorable entity to pursue. In addition, they carry genetic information and other biomolecules, which could potentially serve as biomarkers or even therapeutic targets [32]. Furthermore, several studies have demonstrated that aberrant expression of exosomal noncoding RNAs, such as miRNAs, lncRNAs, and circRNAs, is involved in the regulation of various cellular mechanisms in tumorigenesis. Therefore, these exosomal noncoding RNAs could serve as potential diagnostic, prognostic, and predictive markers [33].

In this review, we summarize the findings of the most recent studies exploring the potential of exosomes that were detected in blood samples of CRC patients. This article focuses on exosomal miR, lncRNA, and circRNA as diagnostic, prognostic, and predictive biomarkers in CRC. In addition, we discuss the advantages and challenges of exosomes as a liquid biopsy target.

## 2. Exosome Biogenesis and Function in CRC

Exosomes are phospholipid bilayer nanovesicles, measuring 30–150nm in diameter that contain a range of molecules, including proteins, lipids, and different types of nucleic acid. Various cell types secrete them, including cancerous cells [34,35] and they are present in many biological fluids [36,37,38,39,40,41,42,43,44,45,46,47,48,49]. Endocytic vesicle invagination on the plasma membrane to form an early endosome represents the initiation step of exosomal biogenesis (Figure 1). Multiple intra-luminal vesicles are created within the endosome via the folding of its phospholipid bilayer. This results in the formation of a multi-vesicular body (MVB), which eventually fuses with the plasma membrane to release its contents, namely exosomes [50]. Thereafter, exosomes can enter the recipient cell utilizing various mechanisms, such as binding cell surface receptors, fusion with the plasma membrane, and internalization via endocytosis [51] (Figure 1).

Although several endosomal-sorting complex required for transport (ESCRT)-independent mechanisms have been identified, exosomal packaging and MVB transportation are primarily regulated via ESCRT and its accessory proteins (Alix, HSC70, HSP90β, and TSG101). These are the “exosomal marker proteins”, since they are expected to be detected on exosomes irrespective of their cell of origin [52,53,54,55]. Other commonly found transmembrane proteins include cluster of differentiation 9 (CD9), CD63, and CD81 [56,57]. In addition to these, exosomes comprise of several types of proteins, including, but not limited to, heat shock proteins (Hsp70), major histocompatibility complex-II, Ras-related protein, signal transduction proteins, transport proteins (annexins, flotillin-1, GPI-anchored proteins, tyrosine kinases), cytoskeleton proteins (actin, tubulin, cofilin), and enzymes (phospholipases) [58,59,60,61,62]. In addition, they are enriched in lipid-rafts, including cholesterol, ceramide, sphingolipids, and phosphoglycerides [32,63]. Furthermore, exosomes contain various types of nucleic acids, including dsDNA, mRNA, miRNA, lncRNA, and circRNA [32].

Exosomes that are secreted by CRC cell lines in vitro or by tissues in vivo have several functions in cancer elicited by its content. They can promote tumor growth [64,65,66], immune evasion [67,68], angiogenesis [69,70,71], metastasis [69,72,73,74,75,76], chemoresistance [76,77,78,79], and inducing endothelial-mesenchymal transition (EMT) in recipient cells [73,74,80,81] (Figure 1).

## 3. CRC Derived Exosomes

Patients with colorectal adenocarcinoma and high-grade adenoma have presented with a significantly higher number of plasma derived exosomes than participants with low-risk lesions (hyperplastic polyps and low-grade adenoma) or healthy controls. In addition, there was significant correlation between the number of exosomes and the size of representative lesion, the number of lesions, and the total volume of the lesions [82]. These findings present a potential screening method for the detection of high-risk individuals. The content of the exosomes has been studied as a potential biomarker for CRC.

### 3.1. Exosomal miRNA in CRC Patients 

miRNAs are small non-coding single-stranded RNAs that are 19-24 nucleotides long. They silence genes post-transcriptionally via binding to the 3′-untranslated regions of mRNAs. Free-circulating and exosomal miRNAs have been found to be dysregulated in cancers and their potential as biomarkers is being explored. Their stability makes them useful biomarkers for study in retrospectively collected samples [83]. As for exosomal miRNA content, it is significantly higher in cancer exosomes in comparison to that of normal exosomes. This is primarily due to the higher content of RNA-induced silencing complex (RISC)-loading complex that converts pre-miRNA to mature-miRNA [84].

#### 3.1.1. Diagnostic Biomarkers

Ostenfeld et al. [85] found eight dysregulated miRs in CRC tissue; miR-16, miR-23b-3p, miR-27b-3p, miR-30b, and miR-30c were downregulated, while miR-23a-3p, miR-27a-3p, and miR-222-3p were upregulated when compared to normal mucosa. However, all of them were upregulated in EpCAM+-exosomes in the plasma of CRC patients when compared to healthy controls (HCs) (Table 1). The levels of these exosomal miRs decreased post-surgery, which indicates they are of tumor origin. Exosomes can be utilized by the tumor tissue to dispose of tumor-suppressor miRNAs to elucidate the dysregulation discrepancy between CRC tissue and exosomes (low in tissue, high in exosome), and they can create pre-metastatic niches in distant areas by loading exosomes with oncogene miRs (high in both tissue and exosome) [86]. Furthermore, Li et al. [87] have demonstrated the downregulation of miR-96 and miR-149 in tissue and plasma from CRC patients as well as in GPC1+ exosomes in plasma from CRC patients in comparison to their healthy counterparts. These levels normalized two months after surgical intervention. Moreover, the overexpression of these miRs significantly increased cell apoptosis, while it decreased cell proliferation as well GPC1 expression in CRC cell lines, plasma of mice bearing these CRC cell line tumors, and xenograft tumors. Exosomal miR-92 was also discovered to be significantly downregulated in CRC patients when compared to those with colorectal adenoma and other non-cancerous (NC) lesions. It was significantly decreased in patients with high-grade intraepithelial neoplasia than in patients with NC lesions [88]. On the other hand, studies have shown that exosomal miR-17, miR-18a, miR-18b, miR-181a, miR-125a, and miR320c are significantly upregulated in the plasma of CRC patients [89,90]. The last two miRs were especially overexpressed in early stage CRC. In differentiating CRC patients from HCs, the addition of miR-125a to CEA resulted in a better predictive model than that of CEA alone [89].

In other studies, several miRs were upregulated, including let-7a, miR-1229, miR-1246, miR-150, miR-21, miR-223, miR-23a, and miR-301a when compared to HCs [91,93]. They were significantly downregulated post-surgical intervention. The sensitivities of miR-21, miR-150, let-7a, miR-223, miR-1224-5p, miR-1229, miR-1246, and miR-23a were 61.4%, 55.7%, 50.0%, 46.6%, 31.8%, 22.7%, 95.5%, and 92.0%, respectively. All but one miR had better sensitivity than CEA and CA19-9, which were 30.7% and 16.0%, respectively [93]. Furthermore, miR-320d was found to significantly distinguish patients with metastatic CRC from those who were non-metastatic with a sensitivity of 62% and a specificity of 64.7%. When it was combined with CEA, the sensitivity and specificity rose to 63.3% and 91.3%, respectively [92]. In summary, these miRs could be utilized as potential diagnostic markers of CRC, whereby some are diagnostic of early stages, while others of distant metastatic disease.

#### 3.1.2. Prognostic Biomarkers

Teng et al. [102] have detected significantly higher levels of miR-193a in the exosomes of metastatic CRC cell lines and plasma of CRC patients with liver metastasis. Cell cycle assessment showed that miR-193a causes G1 phase cell cycle arrest. In addition, its overexpression inhibits cell proliferation by targeting Caprin1, which is represented by a decrease in both Caprin 1 mRNA and protein expression. As a result, CCND2 and c-MYC are also downregulated, since they are downstream of Caprin1. Moreover, this was also demonstrated in in vivo studies, whereby the overexpression of miR-193a resulted in a significant increase in survival of CRC bearing mice. As to the cells’ mechanism of dispensing of this tumor-suppressive miR into exosomes, it was via the major vault protein (MVP), since the knockout of MVP resulted in the accumulation of the miR in the cells instead of the exosomes. Another study that was conducted by Liu et al. [100] demonstrated a significant downregulation of miR-4772-3p in patients with recurrent CRC. The reduced expression was correlated with risk of recurrence and death. Standing alone, this miR had a sensitivity of 78.6% and specificity of 77.1% in distinguishing recurrent from non-recurrent patients. Moreover, miR-4772 was a better predictor for recurrence than tumor location and lymph node metastasis. Yet another downregulated miR in CRC tissue and serum exosomes of CRC patients was miR-6869. A significant association was observed between toll-like receptor 4 (TLR4) and this miR. The overexpression of miR-6869 inhibited cell proliferation and nuclear translocation of phosphorylated NF-κB/p65, reduced inflammatory cytokines IL-6 and TNF-α, and promoted apoptosis. Hence, the tumor-suppressive role of miR-6869 was via regulating the TLR4/NF-κB signaling pathway. Furthermore, poor three-year survival rate was observed in CRC patients with low levels of this serum exosomal miR [94].

Exosomal miR-1229 was found to be significantly upregulated in the CRC cell lines and serum exosomes from CRC patients, which was associated with tumor size, stage, lymphatic metastasis, and poor survival. Moreover, these miR levels decreased post-surgery. The overexpression of miR-1229 promoted tubulogenesis of human umbilical vein endothelial cells (HUVECS) via inhibiting its target protein Homeodomain-interacting protein kinase 2. The inhibition of miR-1229 in xenograft mouse model resulted in an inhibitory effect on tumor growth and angiogenesis [96]. Another set of miRs was identified to be upregulated in CRC tissue, plasma, and exosomes of CRC patients, which include miR-17, miR-18a, miR-18b, miR-19a, miR-19b, miR-20a, miR-20b, and miR-106a. They were also associated with posttreatment relapse [97,101]. In addition, the overexpression of miR-18b resulted in increased cell proliferation [101]. Moreover, a high exosomal miR-19a was associated with a higher tumor stage, serosal invasion, lymphactic permeation, lymph node, and liver metastasis. Further analysis showed that it was an independent risk factor for OS and DFS [97]. Another study that was conducted by Tsukamoto et al. [98] showed that miR-21 was significantly high in primary tumor, liver metastasis tissues, and plasma exosomes of CRC patients. This upregulation was associated with tumor stage and liver metastasis. It was a significantly independent factor in predicting poor DFS and OS in stage II and III CRC patients and poor OS in stage IV patients.

Zeng et al. studied miR-25-3p, which was found to be upregulated in CRC cell lines, CRC tissues, and exosomes in CRC patients with metastasis. It forms a pre-metastatic niche via promoting angiogenesis and vascular permeability. The overexpression of miR-25-3p in HUVECs resulted in a decrease in the expression of Krüppel-like factor 2 and Krüppel-like factor 4. Consequently, genes that were downstream of these factors were dysregulated, whereby VEGFR expression increased, while the levels of ZO-1, occludin, and Claudin5 decreased. Furthermore, it was shown that CRC cells could transfer miR-25-3p to endothelial cells via exosomes. In vivo studies showed that exosomal miR-25-3p increased vascular leakiness and promoted liver and lung metastasis in mice with CRC [99]. miR-17, miR-92a-3p, and miR-203 were other upregulated serum exosomal miRs that were associated with metastasis [103,104]. miR-203 was found to be an independent prognostic factor with a poor DFS and OS in patients with advanced stages. Poor DFS in stage III, while poor OS in stages III and IV, were observed. In addition, a group of patients with high levels of exosomal miR-203 had a higher incidence rate of venous invasion, lymph node metastasis, liver and lung metastasis, and peritoneal dissemination. An increase in liver metastasis was also demonstrated in xenograft mouse models. Concerning the poor prognosis, it is due to the incorporation of exosomes carrying this miR into monocytes, which promoted their differentiation to M2 macrophages. THP-1 cells that were co-cultured with miR-203 transfected CRC cell lines had an increase of CD163 expression (M2 marker) and a decrease in CXCL10 (M1 marker) [104].

Furthermore, Cheng et al. [95] studied sphere-derived colorectal cancer stem cells (CRCSC) from HCT15, HT29, and murine CT26 cells. miR-146a was found to be overly expressed in these cell lines. Moreover, miR-146a-loaded exosomes from these sphere-derived cancer stem cells were associated with a downregulation of the Numb protein in the recipient cells. Upon the introduction of an antagomiR against miR-146a, a decrease in the expression of this miR and intestinal stem cell genes (CD44, ASCL2, CD166), spheroid formation impairment, tumorgenicity, and an increase in the expression of the differentiated gene CK20 and Numb protein levels were observed. Furthermore, CRC patients with a high amount of enriched-miR146a exosomes in serum demonstrated higher miR-146aHigh/NumbLow CRCSC traits. In addition, neutrophils were in abundance, while the CD8+ T cells were scarce. All in all, these numerous miRs have prognostic potential, even with some being TNM stage specific.

#### 3.1.3. Predictive Biomarkers

The CRC cell lines and serum exosomes of chemo-resistant patients to 5-FU and oxaliplatin have demonstrated an upregulation of miR-21, miR-1246, miR-1229, and miR-96 when compared to their respective chemo-sensitive counterparts. GO analysis and KEGG pathway analysis have shown that these miRs are enriched in several pathways, including PI3K-Akt, FoxO, and autophagy. Furthermore, this panel of exosomal miRNAs could significantly distinguish between chemo-sensitive and resistant groups, with a sensitivity of 78% and specificity of 88.9%. Therefore, this panel of exosomal miRs could be a potential predictive marker in CRC patients [105].

#### 3.1.4. Diagnostic and Prognostic Biomarkers

A significant upregulation of exosomal miR-27a and miR-130a was detected in CRC cell lines and the plasma of CRC patients. It was correlated with tumor grade and stage as well as poor OS. These miRs were significantly downregulated post-surgery, which indicates that they were tumor derived. In differentiating HCs from CRC patients, miR-27a had a sensitivity of 81.82% and a specificity of 90.91%, while miR-130a had a sensitivity of 69.32% and specificity of 100%. These miRs combined together had a sensitivity of 85.23% and specificity of 90.91%. Moreover, these exosomal miRs were significantly higher in plasma of CRC patients than HCs. They have also shown potential to differentiate even early stage tumors (stage 1) from HCs. miR-27a had a sensitivity of 75% and specificity of 77.5%, while miR-130a had a sensitivity of 82.5% and specificity of 75%. These miRs combined together had a sensitivity of 82.5% and specificity of 75%. Furthermore, it was shown that they could promote CRC progression via the Wnt/β-catenin and TGFβ pathways, for miR-27a targets include RXRa, SMAD2, SMAD4, and SFRP1, while Nkd2 is a target of miR-130a. Moreover, the inhibition of miR-27a significantly decreased the expression of CD44, TCF4, cyclin D1, c-myc, SMAD2, and SMAD4. In addition, the inhibition of miR-130a decreased the expression of the same proteins as miR-27a, except for SMAD2 and SMAD4 [107].

Liu et al. [109] found that miR-221 was overexpressed in CRC tissues, plasma, and exosomes of CRC patients, which was associated with tumor size, grade, stage, and lymph node metastasis. Patients with upregulated miR-221 in tissues had a survival rate, local recurrence rate, and metastasis rate of 59.32%, 62.25%, and 34.75%, respectively, while the numbers for those with downregulated miR-221 were 92.50%, 25.00%, and 15.00%, respectively. As for those with enriched-miR-221 exosomes had survival rate, local recurrence rate, and distant metastasis rate of 62.39%, 64.96%, and 34.19%, respectively, while the numbers for those with downregulation were 82.93%, 26.83%, and 17.07%, respectively. Furthermore, studies have detected a significant increase in serum exosomal miR-6803 and a significant decrease in exosomal miR548c in CRC patients, especially those at a later stage and those with liver metastasis. They were found to be significant independent factors in predicting poor DFS and OS [106,108].

Interestingly, unlike the results of Ogata-Kawata et al. [93], serum exosomal miR-150 was significantly downregulated in the CRC cohort that was studied by Zou et al. [110]. In addition, post-operative samples detected an upregulation of this miR. The decreased levels were correlated with advanced stage, positive lymph node metastasis, and poor OS and DFS. In addition, miR-150 differentiated CRC patients from healthy individuals with a sensitivity of 81% and specificity of 76.1%. Meanwhile, CEA alone had a sensitivity of 78.2% and specificity of 75%. However, a combination of both markers was the best, whereby sensitivity and specificity increased to 87.2% and 83.3%, respectively. Moreover, this miR’s target was identified as ZEB1. Therefore, exosomal miRs 27a, 130a, 150, 221, 548c, and 6803 could play a major role in the detection and prognosis of CRC.

#### 3.1.5. Prognostic and Predictive Biomarkers

Ren et al. [111] reported a significant upregulation of miR-196b in CRC tissues and the serum exosomes of CRC patients, which correlated with tumor stage, metastasis category, and poor OS. In vitro, the overexpression of miR-196b resulted in enhanced spheroid formation ability, increased fraction of side population cells, stem cell factors, and mitochondrial potential. In addition, it suppressed apoptosis when CRC cells were treated with 5-FU, whereby there was an increase in anti-apoptotic proteins Bcl-2 and Bcl-xL, and a decrease in caspase-3 or caspase-9. However, the cells were re-sensitized to the drug upon the suppression of miR-196b. Moreover, the overexpression of miR-196b decreased the mRNA and protein levels of SOCS1 and SOCS3, but it increased stem cell factors, such as NANOG, BMI-1, OCT4, and SOX2. Therefore, the activation of the STAT3 signaling pathway via miR-196b’s inhibition of the negative regulators SOCS1 and SOCS3 seems to be the mechanism by which it promotes stemness and chemoresistance. miR-196b seems to be a potential prognostic and predictive marker of CRC. In another study, the exosomal miR-125b levels were significantly high in CRC patients, especially in those resistant to mFOLFOX6‑based chemotherapy. It was associated with worse PFS than in patients with low levels of this miR. The miR-125b levels were significantly lower in patients with partial response to the treatment in comparison to levels prior to treatment. The patients with stable disease had no change in miR-125b levels while patients with progressive disease had significantly higher levels post-treatment. Significantly different levels of miR-125b were observed between the groups as early as one-month post-treatment initiation [112]. These results demonstrate the prognostic and predictive potential of these miRs in CRC.

### 3.2. Exosomal IncRNA in CRC patients 

lncRNAs are transcripts that are made up of 200 nucleotides or more [113]. They regulate several biological processes, since they interact with chromatin-modifying proteins and transcription factors. They are implicated in tumorigenesis, progression, and metastasis in several cancers and they are currently studied as potential diagnostic and prognostic factors [114].

#### 3.2.1. Diagnostic Biomarkers

Barbagallo et al. [115] have shown that urothelial carcinoma associated 1 (UCA1) was upregulated in CRC tissues, while downregulated in serum exosomes of CRC patients in comparison to HCs. The sensitivity and specificity of UCA1 levels in differentiating CRC patients from HCs were 100% and 43%, respectively. Moreover, it was demonstrated that UCA1 can be controlled via the binding of CCAAT/enhancer-binding protein beta (CEPB) onto its binding site upstream of UCA1. There was a significant linear correlation between CEPB and UCA1 in CRC tissues, which suggests that CEPB might be a transcription activator. Furthermore, the overexpression of UCA1 in CRC cells resulted in a downregulation of miR-135a, miR-143, miR-214, and miR-1271, and an increase of mRNAs ANLN, BIRC5, IPO7, KIF2A, and KIF23. Moreover, experiments have shown that UCA1 can bind those miRs and mRNAs. They prevent miRNA binding to degrade these target mRNAs via interacting with mRNA 3′-UTRs, which promotes CRC progression. Functional enrichment analysis has implicated UCA1′s involvement in cellular migration (Table 2). As for the asymmetric distribution of UCA1, it is hypothesized that tumor cells limit the secretion of UCA1 via exosomes to retain UCA1, which suggests that it has a crucial oncogenic function in CRC progression.

Studies have demonstrated a significant overexpression of LNCV6_116109, LNCV6_98390, LNCV6_38772, LNCV_108266, LNCV6_84003, LNCV6_98602, and Colon cancer-associated transcript 2 (CCAT2) in CRC tissue, serum levels, and exosomes of CRC patients when compared to HC [116,117]. The levels of all the aforementioned lncRNAs, with the exception of CCAT2, were significantly altered in each stage [117]. Meanwhile, the higher expression of CCAT2 was associated with local invasion and lymph node metastasis. In addition, there was a significant decrease in the CCAT2 serum levels post-operatively [116]. Collectively, these lncRNAs could serve as diagnostic markers of CRC.

#### 3.2.2. Diagnostic and Prognostic Biomarkers

Liang et al. [118] have shown that ribonuclease P RNA component H1 (RPPH1) was significantly upregulated in CRC tissues and exosomes in CRC patients’ plasma, which was associated with advanced stages, poor OS, and poor DFS. This is explained by the findings that RPPH1 promoted CRC metastasis in vitro and in vivo. RPPH1′s interaction with β-III tubulin (TUBB3) prevents its ubiquitination and induces EMT. Its overexpression in CRC cells have demonstrated an increase in vimentin and N-cadherin with a decrease in E-cadherin mRNA and expression levels. Furthermore, RPHH1 promoted CRC cell proliferation and metastasis via the transportation of RPPH1-enriched exosomes from CRC cells into macrophages, which resulted in macrophage M2 polarization. Upon the stimulation of macrophages with RPPH1-enriched exosomes, they manifested a CD206high/HLA-DRlow phenotype, elongated cellular morphology, an increase in M2 markers (CCL17, CCL18, CXCL8, IL-10, and TGF-β), and it increased the number of CTCs in comparison to the control group. It was better than CEA and CA199 markers as for the diagnostic value of exosomal RPPH1 in CRC patients’ plasma, and its levels decreased post-surgery.

In a study that was conducted by Liu et al. [113], colorectal neoplasia differentially expressed - h (CRNDE-h) exhibited an upregulation in CRC cell lines and exosomes in the serum of CRC patients in comparison to patients with benign colorectal disease (BCD) and HCs. Moreover, the exosomal CRNDE-h levels were associated with regional lymph node metastasis, distant metastasis, and low OS. A five-year OS rate analysis showed that exosomal CRNDE-h levels was an independent prognostic factor. In addition, they assessed for the prognostic value of combining serum exosomal CRNDE-h and CEA. It showed that the OS rates in descending order were as follows: patients with low exosomal levels of both, followed by low CRNDE-h and high CEA levels, then high exosomal CRNDE-h alone, and the lowest OS rates in patients with both exosomal levels high. Furthermore, CRNDE-h was better than CEA in differentiating the CRC patients from patients with BCD and HCs with a sensitivity of 70.3% vs. 37.16% and a specificity of 94.4% vs. 88.75%. However, combining both markers was superior to either alone. Therefore, it is recommended to combine exosomal CRNDE-h and CEA levels for better detection rates. Furthermore, the downregulation of growth arrest-specific transcript 5 (GAS5) that was associated with an upregulation of miR-221 was demonstrated in CRC tissues, serum and exosomes of CRC patients. These levels correlated with CRC stage, lymph node metastasis, local recurrence rate, and distant metastasis rate. It was found to be an independent prognostic factor of CRC. miR-221 expression decreased and an inhibition in cell proliferation, migration, and invasion was observed when GAS5 was ectopically over expressed in CRC cell lines [109]. Furthermore, Oehme et al. [119] found another downregulated lncRNA in CRC tissue and serum exosomes from CRC patients, namely, HOXA transcript at the distal tip (HOTTIP). Multivariate analysis showed that it is an independent prognostic factor for OS that was specifically associated with poor OS. These studies provide evidence of the potential diagnostic and prognostic roles of lncRNAs CRNDE-h, GAS5, and HOTTIP.

### 3.3. Exosomal circRNA in CRC Patients 

circRNAs are a subset of lncRNAs that undergo backsplicing, whereby a downstream 5′ and an upstream 3′ splice sites are joined [120,121,122]. This circular loop has a high tolerance to exonucleases. circRNAs are more stable and they have longer half-lives than their linear counterparts [123,124,125]. They have several functions, including acting as miRNA sponges and interacting with RNA-binding proteins, to regulate transcription and alternative splicing [122,126,127]. They are investigated as potential markers and therapeutic targets in cancer, since they are involved in initiation and progression [128,129,130].

#### 3.3.1. Prognostic Biomarkers

Studies have shown that hsa-circ-0005100 (circFMN2) and hsa_circ_0067835 (circIFT80) were significantly upregulated in the CRC cell lines, CRC tissues, and exosomes from CRC patients. Higher circFMN2 and circIFT80 expressions were correlated with tumor size, stage, and distant metastasis. Moreover, circFMN2 and circIFT80 knockdown in vitro and in vivo significantly inhibited cell proliferation. There was an increase in G0/G1 phase cells and a decrease in the G2/M phase cells [131,132]. However, circFMN2 knockdown did not have that effect on the human colonic epithelial cell line. Neither circFMN2 knockdown inhibited cell proliferation, nor its overexpression promoted proliferation [131]. This makes circFMN2 a promising therapeutic target, since normal cells were not affected. In addition, circIFT80 knockdown increased CRC cell apoptosis, while circFMN2 knockdown showed no change in CRC cell apoptosis [131,132]. Furthermore, it has been shown that circIFT80 promotes EMT. The invasion and migration of CRC cells significantly increased in the case of circIFT80 overexpression, which was associated with a decrease in E-cadherin expression and an increase in the expression of vimentin and N-cadherin [132] (Table 3).

Experiments have shown that the mechanism of action of circFMN2 and circIFT80 is through the circFMN2/miR-1182/hTERT axis and circIFT80/miR-1236-3p/HOXB7 axis, respectively. The hTERT and HOXB7 expressions were higher in CRC tissues than in normal tissue. In addition, there was an inverse correlation between the expression levels of the circRNAs and their respective miR targets in cell lines and serum exosomes from CRC patients. Furthermore, circFMN2 and circIFT80 knockdown decreased the expressions of hTERT and HOXB7, respectively, which was counteracted via introducing an inhibitor of their miR targets [131,132]. These results demonstrate that exosomal circRNAs circFMN2 and circIFT80 are potential prognostic markers.

#### 3.3.2. Predictive Biomarkers

Hon et al. [133] studied the role of hsa-circ-0000338 in CRC patients that were resistant to 5-FU and oxaliplatin (FOLFOX). The findings include a significant more than two-fold upregulation of exosomal hsa-circ-0000338 in the serum of non-responders than that of responders. Furthermore, hsa-circ-0000338 knockdown increased the viability of HCT116 cells in 5-FU concentrations of ≥60 µg/mL. Interestingly, they investigated whether chemoresistance was transferable from HCT116-R (resistant) cells to HCT116-P (parental) cells. The results showed that HCT116-R exosomes selectively transferred hsa-circ-0000338 into co-cultured HCT116-P cells. These HCT116-P cells demonstrated an increased viability against drug treatment when compared to the control cells. Therefore, exosomal hsa_circ_0000338 could be a predictive biomarker for CRC patients.

#### 3.3.3. Diagnostic and Prognostic Biomarkers

Pan et al. [134] have shown that exosomal hsa-circ-0004771 was upregulated in CRC patients’ serum in comparison to that of patients with benign intestinal disease and HCs. In fact, the exosomal hsa-circ-0004771 levels were 14-fold higher in CRC patients than in HCs. These levels correlated with stage and distant metastasis. The sensitivity and specificity of these levels in differentiating CRC patients from HCs were 80.91% and 82.86%, respectively. Furthermore, these levels significantly decreased post-operatively and upon treating CRC cells (HCT-116 and SW-480) with an exosome blocker, namely, GW4869. These results showed that these exosomal levels of hsa-circ-0004771 were tumor derived and they can serve as a diagnostic biomarker for CRC.

## 4. Advantages of Utilizing Exosomes over CTCs, cfDNA/ctDNA, and cf-miR

The tumor derived exosomes represent the characteristics of their parent tumor cells, since they carry various molecules, such as proteins and miRNA profiles, amongst others, which affect their tumorigenic potential [135]. In fact, several studies have demonstrated that the biomarkers detected within exosomes highly correlate with the tissue analysis. Moreover, since a lipid bilayer membrane surrounds these molecules, they are protected from being degraded while in circulation. Therefore, after the collection of exosomes, there are numerous diverse downstream analyses that could be done from these vesicles alone. In addition, exosomes may export chemotherapeutic drugs, which makes them ideal for monitoring therapeutic effect and detecting drug resistance prior to relapse [136]. Furthermore, these vesicles are relatively stable at room temperature, which eliminates concerns of potential variability with specimen collection and storage [135].

On the other hand, CTCs and cfDNA/ctDNA are delicate, unstable, and have short half-lives, which requires quick processing post blood sample collection. Unlike in the case of exosomes, it is not possible to long-term bio-bank these blood samples. Moreover, detecting CTCs and cfDNA/ctDNA is another major limitation, since they are rare across studied tumors and even more so in patients with low tumor burden. In addition, most of the cfDNA in the blood is not from tumor cell origin, which makes the detection of ctDNA even harder and limits the number of experiments possible. Additionally, we cannot analyze the RNA transcriptome and proteome of the tumor via ctDNA [136]. Furthermore, despite the marker-based enrichment method of CTCs, they are phenotypically heterogeneous, which limits its effectiveness, and markers could be downregulated during different stages of cancer, such as during EMT [137]. As for cf-miRs, they are prone to contamination from several sources, including cell-derived miRNA from puncture site and lysed red blood cells, platelets, and leucocytes from circulation. In addition, they lack tumor-specificity, whereas tumor-derived exosomal miRs would be highly specific [138].

## 5. Limitations and Challenges of Using Exosomes

Ultracentrifugation is the gold standard for exosome isolation, which happens to be labor-intensive, time-consuming, requires large amount of starting material, expensive equipment, and is not efficient for high-throughput assays [139,140]. Despite the availability of commercial kits forexosome isolation, they are not standardized and result in low purity. In addition, they utilize buffers that digest exosome membranes, which limits qualitative and quantitative analyses, since intact particle collection is not possible. Moreover, ultracentrifugation might result in particle clumps, which limits characterizing single vesicles [140]. In addition, their small size and composition add to the difficulty of developing a golden standard for quantification [141].

## 6. Future Perspectives

One primary focus ought to be the detection of tumor-derived exosomes. Some studies have already demonstrated a significantly higher abundance of EpCAM+, GPC1+, and CD147+ exosomes in the tissue and plasma/serum of CRC patients when compared to their healthy counterparts [85,87,142]. Moreover, Xiao et al. [143] found that the 5-FU resistant CRC cells (HCT-8/5-FU) secreted TAG72-rich exosomes significantly more than normal colon cells. In addition, CK19-rich exosomes and CA125-rich exosomes were significantly higher in normal colon cells and metastatic CRC cells, respectively. Furthermore, there is a need to develop approaches that can purify exosomes in a quicker and more sensitive manner, which ought to also be standardized. This would aid in receiving FDA approval for the utilization of exosomes in CRC patient care. It is worth noting that three exosomal RNAs that are associated with high-grade prostate cancer are assessed via an FDA approved test, namely, the ExoDx™ Prostate (IntelliScore) test. Currently high grade versus low grade prostate cancer is distinguished utilizing this test alongside PSA [144].

## 7. Conclusions

Accumulating evidence suggests that exosomal noncoding RNA content, including miRs, lncRNAs, and circRNAs, have a pivotal role in oncogenesis and resistance to therapy in CRC. Exosomes that were collected from blood samples have demonstrated diagnostic potential in early stages of the disease, prognostic potential that is TNM stage specific, and predictive potential to the most common chemotherapeutic drug and combination regimen in CRC, namely, 5-FU and FOLFOX. Moreover, exosomal noncoding RNAs have been shown to have a higher sensitivity and specificity than CEA and CA19-9 alone. Nevertheless, a combination of these blood-based markers have resulted in sensitivities and specificities that have reached up to 90%. The data collected are promising and further studies will help increase our understanding of exosomes and their potential value as cancer biomarkers despite the technical challenges limiting their clinical application.

## Figures and Tables

**Figure 1 ijms-21-01398-f001:**
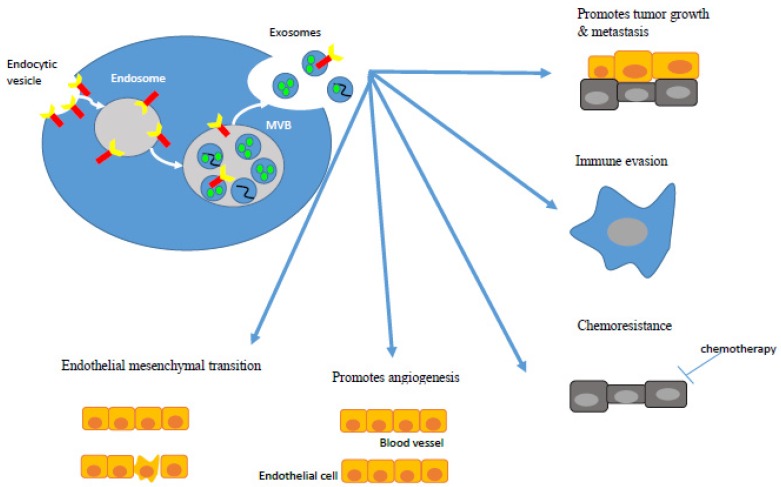
Exosome biogenesis and role in colorectal cancer (CRC).

**Table 1 ijms-21-01398-t001:** Summary of exosomal miRNA studies in CRC patients.

miRNA	Number of CRC Patients*	Sample Type	Exosome Isolation Method	Expression	Type of Biomarker	Refs
miR-16, miR-23a-3p, miR-23b-3p, miR-27a-3p, miR-27b-3p, miR-30b, miR-30c, and miR-222	13	Serum and plasma	Ultracentrifugation	Upregulated	Diagnostic	[85]
miR-96 and miR-149	102	plasma	ExoCap^TM^ exosome isolation and enrichment kit	downregulated	Diagnostic	[87]
miR-23a and miR-301a	25	serum	ExoQuick	upregulated	Diagnostic	[91]
miR-17, miR-181a, miR-18a, and miR-18b	18	plasma	ExoQuick	upregulated	diagnostic	[90]
miR-320d	142	serum	ultracentrifugation	upregulated	Diagnostic	[92]
miR-125a and miR-320c	50	plasma	ExoQuick	upregulated	Diagnostic	[89]
miR-92b	40	plasma	ultracentrifugation	downregulated	Diagnostic	[88]
let-7a, miR-1229, miR-1246, miR-150, miR-21, miR-223, and miR-23a	88	serum	ultracentrifugation	upregulated	Diagnostic	[93]
miR-6869	142	serum	Invitrogen Total Exosome Isolation Kit	downregulated	Prognostic	[94]
miR-146a	53	serum	Ultracentrifugation, ExoQuick	upregulated	Prognostic	[95]
miR-1229	97	serum	ultracentrifugation	upregulated	Prognostic	[96]
miR-19a	227	serum	Invitrogen Total Exosome Isolation Kit	upregulated	Prognostic	[97]
miR-21	326	plasma	ultracentrifugation	Upregulated	Prognostic	[98]
miR-25-3p	75	serum	ultracentrifugation	upregulated	Prognostic	[99]
miR-4772-3p	84	serum	ExoQuick	downregulated	Prognostic	[100]
miR-17, miR-18a, miR-18b, miR-19a, miR-19b, miR-20a, miR-20b and miR-106a	100	plasma	ExoQuick	upregulated	Prognostic	[101]
miR-193a	25	plasma	Ultracentrifugation, exoEasy Maxi Kit	upregulated	Prognostic	[102]
miR-17 and miR-92a-3p	29	serum	Ultracentrifugation and qEV Size Exclusion Columns	upregulated	Prognostic	[103]
miR-203	240	serum	ultracentrifugation	upregulated	Prognostic	[104]
miR-21, miR-1246, miR-1229, and miR-96	43	serum	Ultracentrifugation, Invitrogen Total Exosome Isolation Kit, western blot	upregulated	Predictive	[105]
miR-548c	108	serum	Invitrogen Total Exosome Isolation Kit	downregulated	Diagnostic and Prognostic	[106]
miR-27a and miR-130a	369	plasma	Invitrogen Total Exosome Isolation Kit	upregulated	Diagnostic and Prognostic	[107]
miR-6803	168	serum	Invitrogen Total Exosome Isolation Kit	upregulated	Diagnostic and Prognostic	[108]
miR-221	158	serum	Ultracentrifugation	upregulated	Diagnostic and Prognostic	[109]
miR-150	133	serum	ExoQuick	downregulated	Diagnostic and Prognostic	[110]
miR-196b	150	serum	exoRNeasy Serum/Plasma Maxi Kit	upregulated	Prognostic, and Predictive	[111]
miR-125b	61	plasma	ultracentrifugation	upregulated	Prognostic, and predictive	[112]

* number of CRC patients who had blood samples drawn which were utilized for exosomal analysis.

**Table 2 ijms-21-01398-t002:** Summary of exosomal lncRNA studies in CRC patients.

lncRNA	Number of CRC Patients *	Sample Type	Exosome Isolation Method	Expression	Type of Biomarker	Refs
UCA1	20	serum	Ultracentrifugation, ExoQuick	Downregulated	Diagnostic	[115]
CCAT2	100	serum	ExoQuick	upregulated	Diagnostic	[116]
LNCV6_116109, LNCV6_98390, LNCV6_38772, LNCV_108266, LNCV6_84003, LNCV6_98602	50	plasma	Ultracentrifugation	upregulated	Diagnostic	[117]
RPPH1	52	plasma	Ultracentrifugation, SBI	upregulated	Diagnostic And prognostic	[118]
CRNDE-h	148	serum	ExoQuick	Upregulated	Diagnostic and prognostic	[113]
GAS5	158	Serum	Ultracentrifugation	Downregulated	Diagnostic and Prognostic	[109]
HOTTIP	100	serum	Exosome isolation kit (ThermoFisher Scientific^®^, Cat-Nr.: 4478360)	Downregulated	Diagnostic and Prognostic	[119]

* number of CRC patients who had blood samples drawn which were utilized for exosomal analysis.

**Table 3 ijms-21-01398-t003:** Summary of exosomal circRNA studies in CRC patients.

circRNA	Number of CRC Patients *	Sample Type	Exosome Isolation Method	Expression	Type of Biomarker	Refs
hsa-circ-0005100 (circFMN2)	35	serum	ultracentrifugation	Upregulated	Prognostic	[131]
hsa-circ-0067835 (circIFT80)	58	plasma	ultracentrifugation	Upregulated	Prognostic	[132]
hsa-circ-0000338	17	serum	QIAGEN exoRNeasy Midi Kit, ultracentrifugation	Upregulated	Predictive	[133]
hsa-circ-0004771	135	serum	Invitrogen^™^ Total Exosome Isolation Kits	Upregulated	Diagnostic And Prognostic	[134]

* number of CRC patients who had blood samples drawn which were utilized for exosomal analysis.

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
