# Peer review of "Exosomal Non Coding RNA in LIQUID Biopsies as a Promising Biomarker for Colorectal Cancer"

_ijms, 2020, doi:10.3390/ijms21041398_

Round 1

Reviewer 1 Report

The paper entitled “Exosomal non coding RNA in liquid biopsies as a promising biomarker for colorectal cancer” by Baassiri et al. is a timely review that summarize the diagnostic, prognostic and predictive roles of non-coding RNAs in colorectal cancer and their potential as biomarkers in liquid biopsy due to their presence and transport in exosomes.

Overall, the article is well written, and it provides detailed and precise information on the main non-coding RNAs categorized on the basis of their main diagnostic, prognostic and predictive meaning.

In particular, some points need to be reviewed as detailed below.

Line 22: the sentence, as it is formulated, states that 5-FU and FOLFOX are drugs used in the treatment of CRC: this is misleading as FOLFOX is not a single drug but a therapeutic regimen based on the combination of more substances. Line 458: the same comment as above. The title of paragraph 3.1.4 (line 232) should be “Diagnostic and prognostic biomarkers” Lines 295 – 330: here the distinction between diagnostic and prognostic biomarkers is rather arbitrary. In fact, RPPH1 (diagnostic biomarker) is described to be associated with poor OS and DFS. Conversely, UCA1 is identified as a prognostic biomarker, but no relation of its expression to the outcome in patients is mentioned. Lines 364 – 394: see the previous comment. Here the two circRNAs described are both correlated with tumour stage and distant metastasis, therefore it is not straightforward why hsa-circ0004771 is described as diagnostic whereas circFMN2 and circIFT80 as prognostic biomarkers. Line 432, paragraph 4. Limitations and challenges. It would be appropriate to add here some general comments on the commercial kits for exosomes isolation and analysis, stressing their advantages, drawbacks and potential. Figure 1 has poor resolution. Some minor English revision is required here and there, some examples not exhaustive: line 167 change “for” with “since” line 202 change “another” with “other” line 292 : for they? Line 300 change “which” with “and”

Author Response

Dear Reviewer,

Thank you for your review of our manuscript entitled “Exosomal non coding RNA in Liquid Biopsies as a promising biomarker for Colorectal Cancer" which we submitted to IJMS.

We have much appreciated your constructive comments and we would like to thank you for giving us the opportunity to submit a revised version. We are pleased to inform you that we have undertaken all the revisions requested and we are submitting here a revised manuscript for your consideration. You will find below a detailed response to each point you have raised.

  1. Line 22: the sentence, as it is formulated, states that 5-FU and FOLFOX are drugs used in the treatment of CRC: this is misleading as FOLFOX is not a single drug but a therapeutic regimen based on the combination of more substances.

Thank you for pointing this out. This was noted in the main text of the manuscript in lines 426-427. However, we agree that this ought to be clarified in the abstract as well. Therefore, the sentence along lines 21-23 was modified as such,

“Moreover, they have prognostic potential that is TNM stage specific and could serve as predictive biomarkers for the most common chemotherapeutic drug and combination regimen in CRC which are 5-FU and FOLFOX, respectively.”

  1. Line 458: the same comment as above.

In accordance with the change made in comment #1, the sentence along lines 498-500 was modified as such,

“Exosomes collected from blood samples have demonstrated diagnostic potential in early stages of the disease, prognostic potential that is TNM stage specific, and predictive potential to the most common chemotherapeutic drug and combination regimen in CRC namely, 5-FU and FOLFOX.”

  1. The title of paragraph 3.1.4 (line 232) should be “Diagnostic and prognostic biomarkers” 

Thank you for the correction. It has been corrected.

  1. Lines 295 – 330: here the distinction between diagnostic and prognostic biomarkers is rather arbitrary. In fact, RPPH1 (diagnostic biomarker) is described to be associated with poor OS and DFS. Conversely, UCA1 is identified as a prognostic biomarker, but no relation of its expression to the outcome in patients is mentioned. 

Thank you for your comment. We apologize for this mistake. It has been corrected by moving

RPPH1 from the diagnostic section to the diagnostic and prognostic section. In addition, we moved UCA1 from the prognostic section to the diagnostic section since we added the following sentence in the paragraph discussing UCA1 (lines 311-312),

“The sensitivity and specificity of UCA1 levels in differentiating CRC patients from HCs were 100% and 43%, respectively.”

Table 2 was modified accordingly.

  1. Lines 364 – 394: see the previous comment. Here the two circRNAs described are both correlated with tumour stage and distant metastasis, therefore it is not straightforward why hsa-circ0004771 is described as diagnostic whereas circFMN2 and circIFT80 as prognostic biomarkers.

Thank you for your comment. We apologize for this mistake. It has been corrected by moving

hsa-circ0004771 from the diagnostic section to the diagnostic and prognostic section.  Table 3 was modified accordingly

  1. Line 432, paragraph 4. Limitations and challenges. It would be appropriate to add here some general comments on the commercial kits for exosomes isolation and analysis, stressing their advantages, drawbacks and potential.

Thank you for your input. In fact, we were referring to commercial kits when we discussed the shortcomings of “other methods” in lines 473-474: “Despite the availability of other methods of exosome isolation, they are not standardized and result in low purity.” However, we clarified it by modifying the sentence and discussing it a bit more. Now it reads as follows,

“Despite the availability of commercial kits for exosome isolation, they are not standardized and result in low purity. In addition, they utilize buffers which digest exosome membranes which limits qualitative and quantitative analyses since intact particle collection is not possible.”

Moreover, since it is the limitations and challenges section, it seems to be appropriate to discuss the drawbacks of these kits rather than their advantages and potential.

  1. Figure 1 has poor resolution.

Thank you for pointing it out. We have replaced it with a high-resolution image.

  1. Some minor English revision is required here and there, some examples not exhaustive: line 167 change “for” with “since” line 202 change “another” with “other” line 292 : for they? Line 300 change “which” with “and”

Thank you for your input, the changes have been made as suggested. As for (previously line 292): “for they”, we had meant “since they”. This change was made as well. Moreover, a few other grammatical and spelling errors were corrected as well.

Reviewer 2 Report

The authors focus on a very interesting topic connected both with liquid biopsy and exosomes in colorectal cancers diagnosis and follow up of the patients. Exosomes are the vesicles of interest in the evaluation of cancer metastasis. The present knowledge suggests that exosomes in cancer cells are directly connected with the carcinogenic process and the ability of cancer cells to spread.

Body fluids samples contain more contaminants, including serum abundant proteins, metabolite and even other derived exosomes. The primary method remains purification by size. This represents a disadvantage, some of the protein fragments are enclosed in urine or serum during the isolation process and therefore maybe directly disturb the results. Additionally, not only exosomes but also other extracellular vesicles such as microparticles and apoptotic bodies carry bioactive molecules.

However, despite the feasibility to collect blood samples without harm to the patients, the liquid biopsies did not replace the traditional paraffin blocks or aspiration biopsies directly from the tumors. The same applies to exosomes.

Author Response

Dear Reviewer,

Thank you for your review of our manuscript entitled “Exosomal non coding RNA in Liquid Biopsies as a promising biomarker for Colorectal Cancer" which we submitted to IJMS.

We have much appreciated your constructive comments and we would like to thank you for giving us the opportunity to submit a revised version. We are pleased to inform you that we have undertaken all the revisions requested and we are submitting here a revised manuscript for your consideration. You will find below a detailed response to each point you have raised.

The authors focus on a very interesting topic connected both with liquid biopsy and exosomes in colorectal cancers diagnosis and follow up of the patients. Exosomes are the vesicles of interest in the evaluation of cancer metastasis. The present knowledge suggests that exosomes in cancer cells are directly connected with the carcinogenic process and the ability of cancer cells to spread.

Body fluids samples contain more contaminants, including serum abundant proteins, metabolite and even other derived exosomes. The primary method remains purification by size. This represents a disadvantage, some of the protein fragments are enclosed in urine or serum during the isolation process and therefore maybe directly disturb the results. Additionally, not only exosomes but also other extracellular vesicles such as microparticles and apoptotic bodies carry bioactive molecules.

However, despite the feasibility to collect blood samples without harm to the patients, the liquid biopsies did not replace the traditional paraffin blocks or aspiration biopsies directly from the tumors. The same applies to exosomes.

Thank you for your comment. Your concerns about the impurities associated with exosome collection are valid. This emphasizes the paramount importance for the need to develop methods which can detect and collect tumor-derived exosomes with high purity. We have mentioned in the manuscript “Some studies have already demonstrated a significantly higher abundance of EpCAM+, GPC1+, and CD147+ exosomes in tissue and plasma/serum of CRC patients when compared to their healthy counterparts” which demonstrates that we are on the right track towards that goal.

Concerning traditional paraffin blocks and aspiration biopsies, they have their own shortcomings where tumor-derived exosomes excel. For instance, the genetic heterogeneity of a tumor could not be represented merely by a biopsy whereas tumor-derived exosomal content could since they are shed by all tumor cells. In addition, patients with cancer need to be frequently followed up to assess response to treatment and to detect relapse early on. Tissue biopsy for follow-up is not a feasible option whereas tumor-derived exosome collection via a blood sample is. Moreover, tissue biopsy acquisition is not even feasible to begin with in certain sites where cancer may arise. The future of patient management may include a combination of both. One needn’t to replace the other. This will depend largely on technological advancements in the foreseeable future.